# Identifying Statements Crucial for Awareness of Interpretive Nonsense to Prevent Communication Breakdowns

**Tomoyuki Maekawa** and **Michita Imai**

Keio University

Yokohama, Japan

{maekawa, michita}@ailab.ics.keio.ac.jp

## Abstract

During remote conversations, communication breakdowns often occur when a listener misses certain statements. Our objective is to prevent such breakdowns by identifying Statements Crucial for Awareness of Interpretive Nonsense (SCAINs). If a listener misses a SCAIN, s/he may interpret subsequent statements differently from the speaker's intended meaning. To identify SCAINs, we adopt a unique approach where we create a dialogue by omitting two consecutive statements from the original dialogue and then generate text to make the following statement more specific. The novelty of the proposed method lies in simulating missing information by processing text with omissions. We validate the effectiveness of SCAINs through evaluation using a dialogue dataset. Furthermore, we demonstrate that SCAINs cannot be identified as merely important statements, highlighting the uniqueness of our proposed method.

## 1 Introduction

Communication breakdowns are sometimes caused by listeners missing speaker's statements in conversations. Listeners tend to miss statements when they are absent-minded or lost in thought. In online conversations, there are other obstacles such as network failures and background noise, which may cause more statements to be missed than in face-to-face conversations.

Problems associated with listeners missing statements are not just the information in the missing statements not being conveyed. The listener's interpretation of other statements may also change because they missed previous statements. Table 1 shows an example of a dialogue between two speakers. With statement 9, speaker A wants to say that *Tokyo* is a very attractive place for her. Suppose a listener missed statements 7 and 8, as shown in Table 2. The listener might interpret statement 9 to mean that *mountainous areas* are attractive to

Table 1: Example of a dialogue. Retrieved from JPesonaChat (Sugiyama et al., 2023). Originally in Japanese; translated into English for inclusion in this paper.

| # | | Statement |
|---|---|---|
| 1 | A: | Hello. How are you? |
| 2 | B: | I'm fine. But I'm busy with my work at the advertising agency. |
| 3 | A: | Thank you. I'm a care worker. |
| 4 | B: | You work as a care worker? That's a tough job, isn't it? |
| 5 | A: | Exactly. Working at an advertising agency must be hard too. By the way, I live in a one-story house in Hokkaido. |
| 6 | B: | You live in Hokkaido? I envy you because that's a place with good food. I'm from Aomori and I like mountainous areas, so I live near the mountains. |
| 7 | A: | There are many delicious foods in Hokkaido. You're from Aomori, so you're close by. But I'd like to live in Tokyo one day. |
| 8 | B: | Tokyo? Tokyo has many gorgeous places, doesn't it? |
| 9 | A: | Because I was born in the countryside, it's a very attractive place. |

speaker A, which contradicts the primary intent of statement 9. Importantly, the listener might not be aware of the missing statements because statement 9 was still interpretable without statements 7 and 8, which could lead to a breakdown in the communication.

Online conversations can be supported by existing techniques, such as dialogue summarization (Chen et al., 2021) and discourse structure visualization (Holmer, 2008). However, previous studies have not considered the change in the interpretation caused by missing statements. Speakers need to be aware that the listener's interpretation may

Table 2: Example of a dialogue with two statements omitted. The interpretation of statement 9 may differ from that in Table 1.

| # | | Statement |
|---|---|---|
| 1 | A: | Hello. How are you? |
| 2 | B: | I'm fine. But I'm busy with my work at the advertising agency. |
| 3 | A: | Thank you. I'm a care worker. |
| 4 | B: | You work as a care worker? That's a tough job, isn't it? |
| 5 | A: | Exactly. Working at an advertising agency must be hard too. By the way, I live in a one-story house in Hokkaido. |
| 6 | B: | You live in Hokkaido? I envy you because that's a place with good food. I'm from Aomori and I like mountainous areas, so I live near the mountains. |
| 9 | A: | Because I was born in the countryside, it's a very attractive place. |

differ from their intended meaning, even though it is difficult to detect that the listener has missed a statement.

Therefore, statements crucial for interpreting other statements need to be detected to prevent communication breakdowns. Specifically, our goal is to identify the Statements Crucial for Awareness of Interpretive Nonsense (SCAINs). The "nonsense" abbreviated in SCAIN has a special meaning. A listener who has not heard SCAINs may interpret a subsequent statement differently from the speaker's primary intent. However, the listener is likely to believe that his/her interpretation is correct.

We assume that SCAINs will be used to develop a conversation support system that is implemented on online meeting tools. Figure 1 shows a diagram of the assumed system. Speech from the speakers is sent to the online meeting server. The transcript of the speech is fed into the SCAIN extractor. The SCAIN is reported to the speakers in the online meeting application so that the speakers are aware of possible interpretations that differ from their intended meaning.

The SCAIN extractor computes the change in interpretation by a listener who missed hearing certain statements. In this sense, SCAINs are different from locally ambiguous statements. Even if a statement is locally ambiguous, the statement is not a SCAIN unless the interpretation changes due to mishearing. The SCAIN extractor picks up

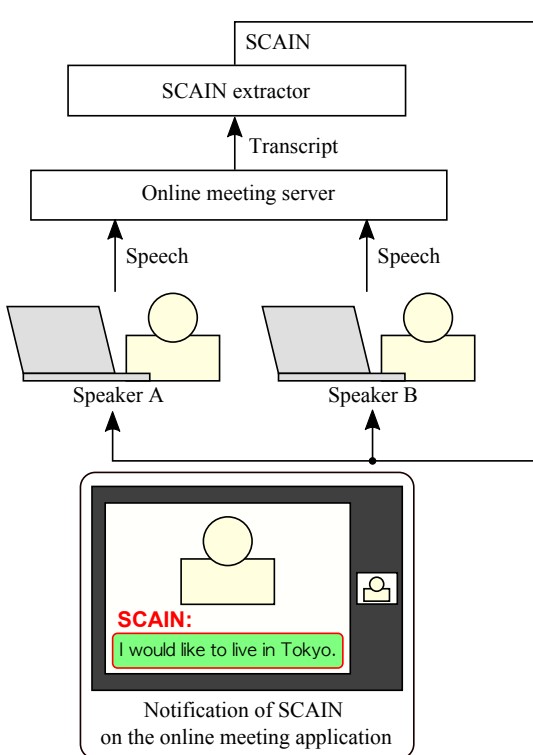

Figure 1: Diagram of the support system for online conversation using SCAINs.

the statements when mishearing may change the interpretation of the subsequent statement, but the listener is not aware of the change, so the conversation support system will alert the speakers to pay attention to the different interpretations.

This paper proposes the concept of SCAIN as a novel idea that aims at mutual understanding through conversations. We explain the method for identifying SCAINs, which uses dialogue with some statements omitted from the original. Then, we evaluate the validity and uniqueness of SCAINs using a dialogue dataset.

The contributions of this paper are two-fold:

- We first propose the method of extracting SCAINs from a dialogue by creating another dialogue assuming that some statements have been missed.

- We validate that SCAINs are indeed crucial to the interpretation of the dialogue and that SCAINs cannot be identified as merely important statements.

## 2 Related Work

### 2.1 Important statements extraction

Conversations can be supported by using techniques to extract important statements. Sentence

importance scoring has been integrated into the process of generating summaries (Moratanch and Chitrakala, 2017). In the context of dialogue summarization, important statements are typically regarded as those containing salient information, named entities, and speaker emotion and intent (Chen et al., 2021). However, in previous methods, scores have been assessed after reading all statements, without considering the possibility of overlooked or missed statements.

## 2.2 Topic shift detection

A conversation can contain multiple topics, and sometimes the topic shifts during the conversation. Detecting the topic shifts is important to promote smooth discussion in online communities (Sun and Loparo, 2019). Xie et al. (2021) introduced a TopicshIft Aware dialoG datasEt (TIAGE) and proposed dialogue models that detect and generate topic-shift responses. However, it is unclear whether topic shifts change the interpretation of the subsequent statement.

## 2.3 Discourse structure and cohesion analyses

Discourse structure and cohesion analyses can also be used for dialog comprehension, as a means of providing the listener with useful information to overcome the missing statements. Discourse structure analysis extracts various kinds of relationships between sentences in a structured way, which allows one to find out the sentences that are crucial for understanding a particular sentence (Webber et al., 2012). Holmer (2008) proposed ChatLine, an application for analyzing the discourse structure of chat transcripts. Connections between statements can be represented visually by using ChatLine. Cohesion analysis refers to the resolution of various types of relationships between words and phrases in a text, such as coreference and bridging anaphora (Ueda et al., 2020). Coreference resolution is useful for dialogue comprehension because pronouns and ellipses are often observed in dialogues (Lin et al., 2016). However, discourse structure and cohesion analyses are not sufficient to simulate the change in interpretation caused by missing statements.

## 2.4 Context dependence

Our approach is to track the change in the interpretation of statements depending on the dialogue context. Some statements in a dialogue are context-dependent in the sense that a listener must consider the previous statements in order to respond

to the statement (Li et al., 2016). Shibata et al. (2023) proposed a method to transform an arbitrary search query into another search query that fits in the dialogue context. However, there is no existing method to generate a natural language text that represents the interpretation of a statement from the perspective of a particular context.

## 3 SCAIN extraction architecture

### 3.1 Overview

The key idea of the SCAIN extraction architecture is to simulate the interpretation of a statement from two different perspectives: a listener who has missed some previous statements and a speaker who has not missed any statements. The listener's perspective is simulated by constructing a dialogue with omissions, while the speaker's perspective is simulated by constructing a dialogue without omissions.

We extract SCAINs by judging whether a pair of consecutive statements are SCAINs or not. Figure 2 shows the overall architecture for extracting SCAINs from a dialogue. We refer to a particular pair of consecutive statements as the candidate statements, except for the first and last statements in the dialogue. The statement following the candidate statements is called the core statement. We take several statements from the original dialogue to construct two shorter dialogues: the complete dialogue and the omitted dialogue. We build a prompt with each dialogue by using a prompt template. A large language model (LLM) generates a more specific statement by rephrasing the core statement in each prompt. Each rephrased statement is embedded into a vector. We compute the similarity score between two embedding vectors. Finally, we compare the similarity score with a predetermined threshold to determine whether the candidate statements are SCAINs or not.

### 3.2 Constructing complete and omitted dialogues

We construct the complete and omitted dialogues for each pair of statements. The complete dialogue consists of the statements from the first statement to the core statement. For example, if statements 7 and 8 are the candidate statements, the complete dialogue is created by concatenating statements 1, 2, 3, 4, 5, 6, 7, 8, and 9. On the other hand, the omitted dialogue consists of the statements from the first statement to the core statement, except

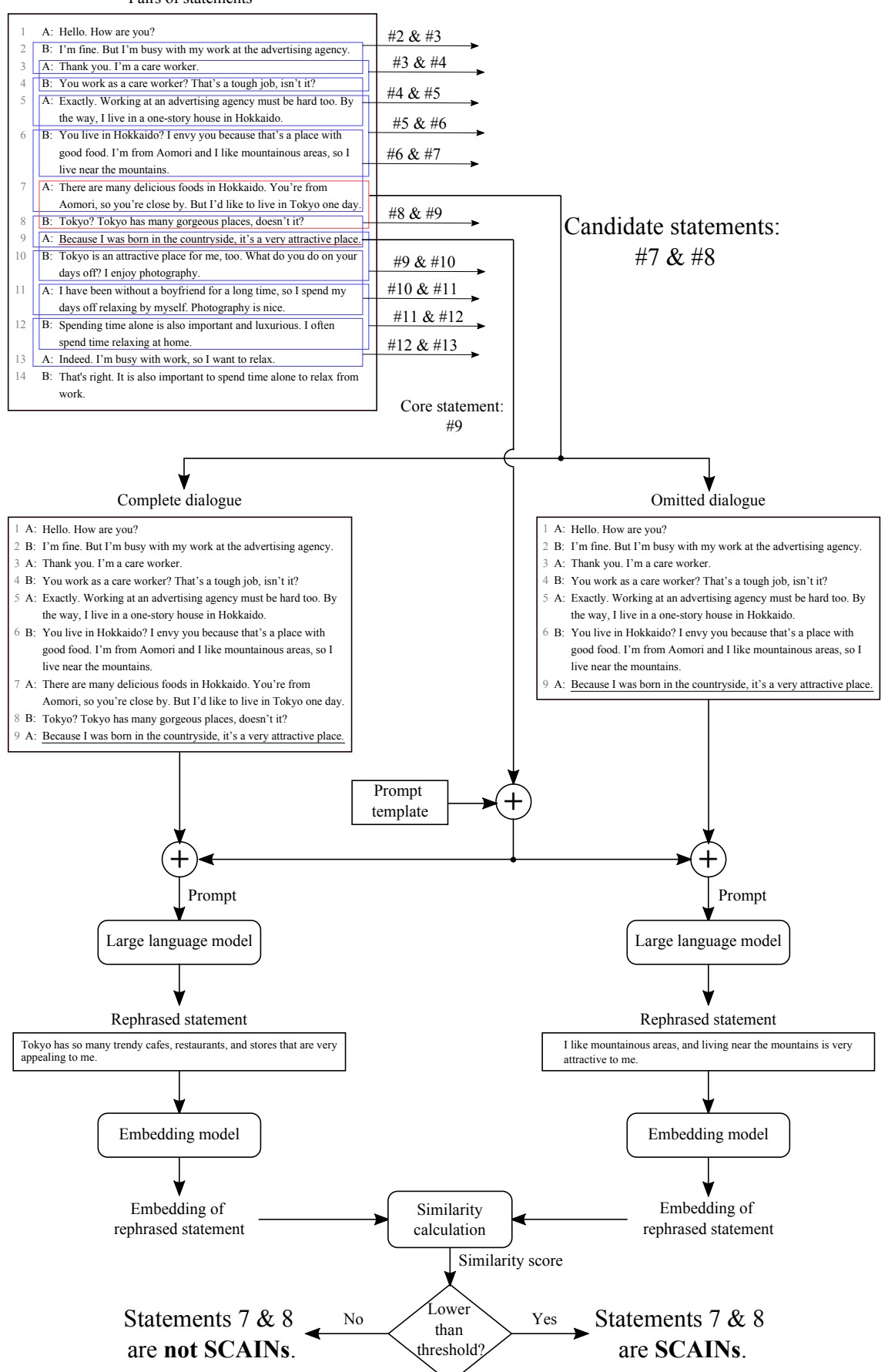

Figure 2: Diagram of the architecture to extract SCAINs from a dialogue.

Table 3: Prompt template to rephrase the core statement. {dialogue} is filled in with either the complete or omitted dialogue. {core statement} is filled in with the core statement.

---

Rephrase a particular statement in a dialogue into a more specific statement.
A dialogue between A and B is written and one of the statements is assigned to be rephrased.
Rephrase the assigned statement into a more specific statement using the words in the dialogue.

# Dialogue
{dialogue}

# Statement to be rephrased
{core statement}

# Rephrased statement

---

for the candidate statements. If we focus on statements 7 and 8, the omitted dialogue is created by concatenating statements 1, 2, 3, 4, 5, 6, and 9. The omitted dialogue simulates the situation where a listener misses the candidate statements, while the complete dialogue simulates the situation where no statements are missed.

### 3.3 Rephrasing the core statement

We obtain the rephrased statements by prompting an LLM. Table 3 shows the prompt template used to build the prompt. The LLM generates a more specific statement under the interpretation on the basis of the context. When the complete dialogue is provided, the LLM takes into account the complete context to rephrase the core statement, so that the rephrased statement represents the primary intent of the speaker. On the other hand, when the omitted dialogue is provided, the LLM does not have access to the candidate statements. In this case, the rephrased statement represents the interpretation of the core statement from the perspective of a listener who has missed the candidate statements.

Our method takes advantage of the LLM's ability to translate a text in the manner specified in the prompt (Brown et al., 2020). It is noteworthy that the LLM never ignores any statements in the prompt, so we need the proposed architecture to simulate the situation where certain statements are missed.

### 3.4 Calculating similarity score

We obtain the embeddings of the rephrased statements by using an embedding model. The embeddings are vectors in the embedding space, where the shorter distance between two vectors indicates that the corresponding texts are more similar. The similarity score between the embeddings is computed as the cosine similarity. We set a threshold and compare the similarity score to the threshold. If the similarity score is lower than the threshold, which means that the interpretation of the core statement from the perspective of the missing listener is significantly different from the primary intent of the speaker, the candidate statements are determined to be SCAINs. Otherwise, missing the candidate statements does not significantly change the interpretation of the core statement, so the candidate statements are determined to be non-SCAINs.

## 4 Evaluation

### 4.1 Overview

We performed two types of evaluations: validity evaluation and uniqueness evaluation. The purpose of the validity evaluation was to show that the SCAINs extracted by our proposed method are significantly crucial for interpreting the subsequent statement. The purpose of the uniqueness evaluation was to show that the SCAINs cannot be extracted by another method, particularly as important statements or topic shifts.

### 4.2 Validity evaluation

#### 4.2.1 Setup

We used 100 dialogues in JPersonaChat (Sugiyama et al., 2023). We applied our extraction method to a total of 952 pairs of statements. Since the dialogues were written in Japanese, we used the Japanese translation of the prompt template. We used gpt-3.5-turbo (Brown et al., 2020) as the LLM and entered the prompt as content for the "user" role. Table 4 lists the parameters set for the LLM. We used text-embedding-ada-002 (Neelakantan et al., 2022) as the embedding model. We set the threshold for the similarity score at 0.85 on the basis of the observation of the rephrased statements. Figure 3 shows the histogram of the similarity scores computed for the pairs of statements. We identified 65 pairs of statements as SCAINs that ended up with similarity scores lower than 0.85. Here, we considered the SCAINs as pairs because we wanted

Table 4: Parameters set for the LLM in the evaluations.

| Parameter | Value |
|---|---|
| Model | "gpt-3.5-turbo" |
| Max tokens | 200 |
| Temperature | 0 |
| Stop words | "\n" |

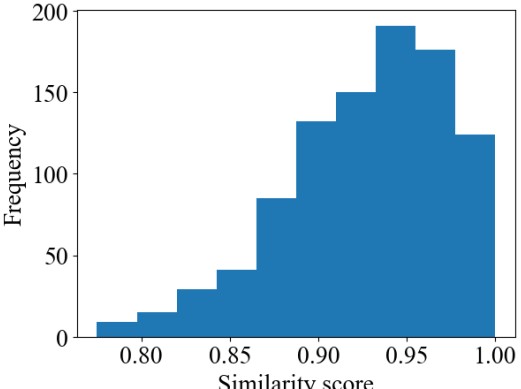

Figure 3: Histogram of the similarity scores.

to analyze the tendency of the dialogues containing the SCAINs.

#### 4.2.2 Conditions

The validity evaluation was conducted with human raters in a within-participant design. In particular, there were two conditions for the pairs of statements to be rated:

- SCAIN condition: We selected SCAINs.

- Non-SCAIN condition: We selected statements that ended up with similarity scores higher than 0.9897. This was done to obtain the same fraction of the statements as the SCAINs with the highest similarity scores.

We set these extreme conditions for the basic analysis to find out tendencies in the dialogues extracted by the proposed method.

#### 4.2.3 Metrics

For each pair of statements, both the complete and the omitted dialogues were presented to the raters. The omitted dialogue was presented as "dialogue 1", and the complete dialogue was presented as "dialogue 2". The raters assessed the interpretation of the core statement in the dialogues by using the following metrics: *clarity* and *suddenness* were assessed for each dialogue and *difference* was assessed for the comparison between the complete

and omitted dialogues. Table 5 lists the details of the metrics.

#### 4.2.4 Procedure

We crowdsourced 30 raters. We created 10 online questionnaires and assigned three raters to each questionnaire. Each questionnaire contained a total of 10 sets of dialogues: five in the SCAIN condition and the other five in the non-SCAIN condition. There was no duplication of dialogues within any of the questionnaires. The dialogues were presented in a random order, and the raters were blinded to the conditions. For each set of dialogues, the raters first read the omitted dialogue and answered the questions for the omitted dialogue. They then read the complete dialogue and answered the questions for the complete dialogue. At this point, we asked the raters not to change the answers for the omitted dialogue to ensure that they were rating the omitted dialogue without reading the complete dialogue. Finally, they answered the question for the comparison between the complete and omitted dialogues.

#### 4.2.5 Prediction of the results

We hypothesized that the SCAINs would significantly affect the interpretation of the subsequent statement. If the SCAINs were omitted from the dialogue, the intent of the core statement would be less clear, the core statement would be perceived as more sudden, and the meaning of the core statement would be different from that of the complete dialogue. In summary, we predicted the following results:

- *Clarity* in the omitted dialogue would be lower in the SCAIN condition than in the non-SCAIN condition.

- *Suddenness* in the omitted dialogue would be higher in the SCAIN condition than in the non-SCAIN condition.

- *Difference* would be higher in the SCAIN condition than in the non-SCAIN condition.

#### 4.2.6 Results

Figure 4 shows the averages and standard deviations of the ratings. Mann-Whitney $U$ tests revealed that *clarity* in the complete dialogue is higher in the SCAIN condition ($U = 9785.0$, $p = 0.030$), *suddenness* in the complete dialogue is lower in the SCAIN condition ($U = 12926.0$, $p = 0.015$), *clarity* in the omitted dialogue is

Table 5: Metrics used in the validity evaluation. These metrics were assessed with 5-point Likert scales. Placeholders marked with {core statement} were filled in with the text of the core statement. The questions were originally in Japanese and were translated into English for inclusion in this paper.

| Metrics for each of the complete and the omitted dailogues | |
|---|---|
| **Metric** | **Question** |
| Clarity | Is the intent of the statement "{core statement}" clear in this context? (1: not at all clear, 5: very clear) |
| Suddenness | Is the statement "{core statement}" sudden in this context? (1: not at all sudden, 5: very sudden) |

| Metric for the comparison between the complete and omitted dialogues | |
|---|---|
| **Metric** | **Question** |
| Difference | Are the meaning of the statement "{core statement}" in dialogue 1 and the meaning of the statement "{core statement}" in dialogue 2 different from each other? (1: completely equal, 5: completely different) |

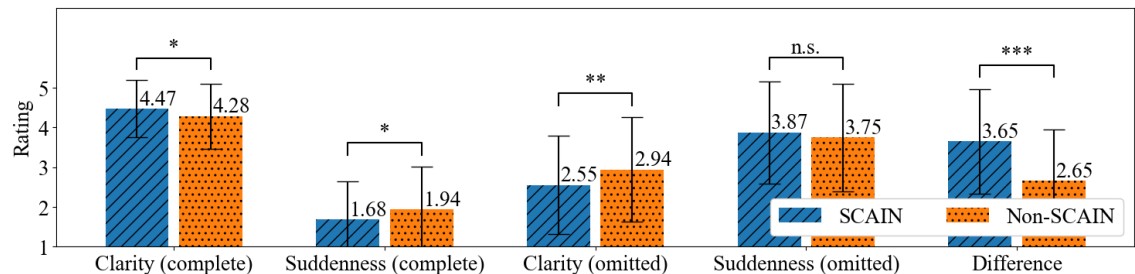

Figure 4: Results of the validity evaluation. Bars represent averages and error bars represent standard deviations. $*p < 0.05$, $**p < 0.01$, $***p < 0.001$.

lower in the SCAIN condition ($U = 13155.5$, $p = 0.009$), *suddenness* in the omitted dialogue is not significantly different between the conditions ($U = 10735.0$, $p = 0.47$), and *difference* is higher in the SCAIN condition ($U = 6721.5$, $p < 0.001$).

### 4.3 Uniqueness evaluation

#### 4.3.1 Setup

The uniqueness experiment consists of the following two comparisons:

- Comparison between SCAINs and important statements

- Comparison between SCAINs and topic shifts

For the comparison between SCAINs and important statements, we used the same 100 dialogues as in the validity evaluation. We extracted SCAINs using the same LLM, embedding model, and parameters as in the validity evaluation. In addition, we extracted important statements by using another prompt template. We numbered the statements

in each dialogue and inserted the dialogue into the prompt template shown in Table 6. See Appendix A for other prompt templates that were considered before the evaluation.

For the comparison between SCAINs and topic shifts, we used the TIAGE test dataset (Xie et al., 2021) annotated on PersonaChat (Zhang et al., 2018). For each statement in the TIAGE dataset, the annotators annotated whether or not there was a topic shift. We extracted SCAINs using the English version of the prompt template while keeping the other settings the same as in the validity experiment.

#### 4.3.2 Conditions

The uniqueness evaluation was performed by classifying the statements in the dialogues from three different aspects. One of the classification aspects was whether the statements were SCAINs or non-SCAINs:

- SCAINs: A statement was identified as a SCAIN if at least one pair containing the state-

Table 6: Prompt template to extract important statements. {dialogue} is filled in with the dialogue.

---

Select the most important statements from the dialogue. A sequence of dialogue is written with numbered statements. Select the two most important statements from the dialogue and write their numbers.

# Format of output
[statement number 1, statement number 2]

# Dialogue
{dialogue}

# Important statements

---

ment ended up with a similarity score lower than 0.85. We obtained a total of 125 SCAINs out of 1252 statements from the JPersonaChat dataset and 84 SCAINs out of 1464 statements from the PersonaChat dataset.

- Non-SCAINs: A statement was otherwise identified as a non-SCAIN. We identified a total of 1127 statements as non-SCAINs in the JPersonaChat dataset and 1380 statements in the PersonaChat dataset as non-SCAINs.

Here, we considered the SCAINs as individual statements because we wanted to analyze the relationship between the classifications of the statements in terms of being SCAINs, important statements, and topic shifts. The number of SCAINs may be less than twice the number of pairs because there are overlapping pairs.

The other classification aspects were the importance and topic shifts. For the comparison between SCAINs and important statements, we classified the statements into important statements and non-important statements:

- Important statements: A statement was identified as important if the statement number was listed in the output of the LLM. We obtained a total of 198 important statements because the LLM did not generate any statement numbers for one of the dialogues.

- Non-important statements: A statement was not otherwise identified as important. We identified a total of 1054 statements as non-important statements.

Table 7: Results of the uniqueness evaluation.

| | Important | Non-important | Total |
|---|---|---|---|
| SCAIN | 25 | 100 | 125 |
| Non-SCAIN | 173 | 954 | 1127 |
| Total | 198 | 1054 | 1252 |

For the comparison between SCAINs and topic shifts, we classified the statements into topic shifts and non-topic shifts:

- Topic shifts: A total of 315 statements in the PersonaChat dataset were annotated as having topic shifts.

- Non-topic shifts: The remaining 1149 statements in the PersonaChat dataset were annotated as having no topic shifts.

### 4.3.3 Prediction of the results
We hypothesized that the SCAINs would capture a unique aspect different from the importance of the statements. Consequently, we predicted that the classification based on the SCAINs would be independent of the classification based on importance and topic shifts.

### 4.3.4 Results
Table 7 shows the frequencies of the classified statements for the comparison between SCAINs and important statements. As a result, 25 SCAINs were classified as important statements, while the remaining 100 SCAINs were classified as non-important statements. A chi-square test showed that the classification based on SCAINs is independent of the classification based on importance ($\chi^2(1) = 1.49, p = 0.22$). Some qualitative results are shown in Appendix B.

Table 8 shows the frequencies of the classified statements for the comparison between SCAINs and topic shifts. Similarly, 18 SCAINs were classified as having topic shifts, while the remaining 66 SCAINs were classified as not having topic shifts. A chi-square test showed that the classification based on SCAINs was also independent of the classification based on topic shifts ($\chi^2(1) = 0.0, p = 1.0$).

## 5 Discussion

The validity of the SCAINs was confirmed in that when the SCAINs were omitted, the intent

Table 8: Results of uniqueness experiment for comparison between SCAINs and topic shifts.

|  | Topic shift | Non-topic shift | Total |
|---|---|---|---|
| SCAIN | 18 | 66 | 84 |
| Non-SCAIN | 297 | 1083 | 1380 |
| Total | 315 | 1149 | 1464 |

of the subsequent statements became unclear and the meaning of the subsequent statements changed. On the other hand, the statements became sudden when the preceding statements were omitted, regardless of whether the preceding statements were the SCAINs or not. The statements may have been perceived as sudden because they were inconsistent with previous statements, not just because the intent of the statement was unclear. In contrast, the intent of the statements became clearer and the statements became less sudden when the SCAINs were provided before the statements. There may have been a context effect caused by the order in which the dialogues were presented. In the SCAIN condition, the raters may have found the complete dialogues relatively clearer and less sudden after rating the omitted dialogues, which were less clear.

If the LLM is informed that some statements have been omitted, it can generate a possible interpretation of the core statement by predicting the content of the omitted statements. However, the generated interpretation is likely to be different from the intended meaning because crucial information for interpreting the core statement has been lost.

The uniqueness of the SCAINs was confirmed by the fact that the SCAINs were identified regardless of the importance of the statements and topic shifts. We found different characteristics for the SCAINs and for the important statements through qualitative analysis. Particularly, the SCAINs had the following characteristics:

- The SCAINs tended to introduce new topics not previously mentioned in the dialogues.

- The statements following the SCAINs tended to contain pronouns and null anaphoras referring to the terms in the preceding SCAINs.

On the other hand, the important statements had the following characteristics:

- The important statements tended to contain information about the speaker's personal attributes.

- The important statements tended to contain the speaker's opinions and/or suggestions.

Note that JPersonaChat consisted of dialogues in which the speakers tried to get to know each other, so most statements contained some kind of information about the speaker's personal attributes. Therefore, it may have been difficult to select only two important statements from a dialogue.

We speculate that there is no significant relationship between the SCAINs and topic shifts because the topic shifts do not necessarily change the interpretation of the subsequent statement. Even if there is a topic shift, the listener may correctly interpret a statement that contains sufficient information about the new topic.

The SCAIN extraction architecture can be applied to a conversation support system. The system will inform the sender of SCAINs which statements are SCAINs during conversations. The sender will pay close attention to verbal and nonverbal responses from the listener. The sender will be able to identify the communication breakdowns earlier and recover from the breakdowns by supplementing the SCAINs with more information. If the speakers do not use the system, the sender will not care if his/her intent is conveyed to the listener and will continue with a one-way conversation. The system will prevent such situations and let the speakers be aware of communication breakdowns.

## 6 Conclusion

In this paper, we proposed the concept of Statements Crucial for Awareness of Interpretive Nonsense (SCAIN) and a method for extracting SCAINs from a dialogue. We confirmed the validity of SCAINs with the results showing that *clarity* in the omitted dialogue was significantly lower in the SCAIN condition and that *difference* was significantly higher in the SCAIN condition. We also confirmed the uniqueness of the SCAINs with the results showing that there was no significant dependence between the classification based on SCAINs and the classification based on importance. In future work, we will develop a conversation support system using SCAINs and evaluate the effectiveness of the system with an experiment in a real environment.

## Limitations

The current research has a limitation in that the results are highly dependent on the LLM used in the architecture. The outputs of LLMs are often biased (OpenAI, 2023), meaning that the rephrased statements do not cover all possible interpretations by the listener, especially if the omitted dialogue does not provide sufficient context. Furthermore, there may be individual differences in the interpretation of a statement, depending on the listener's background and knowledge, which cannot be simulated with a single prompt.

As shown in (OpenAI, 2023), the performance of GPT-4 in Japanese language processing is inferior to that of English. We speculate that GPT-3.5-turbo has a similar tendency. However, we have confirmed that GPT-3.5-turbo shows sufficient accuracy for the task of rephrasing a sentence into a more specific one.

Moreover, the SCAIN extractor requires four forward passes of the LLM for each utterance. Currently, the computation time of the LLM cannot be ignored. However, the inference process of the LLM is becoming more efficient thanks to the progress in computer hardware and LLM research. Therefore, we expect that the SCAINs will be extracted in a much shorter time for real-time applications.

Another limitation is that the current approach to extracting SCAINs focuses only on the pair of statements immediately preceding the core statement. We recognize that there is a need to identify more distant SCAINs. The current study aimed to establish the concept of SCAIN, so we defined the candidate statements as the immediately preceding statements for the first step. We will extend the method to identify more distant SCAINs as an important future work. In addition, there will be an issue of computational cost as we increase the number of patterns of candidate statements. We will be able to predict the change in interpretation of the core statement caused by the omission of more distant statements if we provide sufficient computational resources. This extension will be done when the inference process of the LLM becomes more efficient thanks to research progress.

We predetermined the threshold for identifying SCAINs in the evaluation because we wanted to capture the tendency of SCAINs. The results of the validity experiment do not guarantee that the predetermined value of the threshold is valid. We expect dialogues with intermediate similarities between the SCAINs and Non-SCAINs conditions will fall into intermediate results. The evaluation results show that the proposed method can extract dialogues with significantly different tendencies, at least for extreme conditions. However, the threshold needs to be determined on the basis of a more detailed experiment that examines the relationship between the similarity score and the change in interpretation of the core statement.

Finally, the uniqueness of the SCAINs needs to be evaluated in more diverse ways. In this paper, we compared the classification based on the SCAINs with the classification based on the importance of the statements evaluated by the LLM. There are other approaches, such as discourse structure and cohesion analysis, that need to be compared with SCAIN extraction.

## Ethics Statement

We believe that there is little concern about ethical issues related to our research since we assume that the support system will only display some of the speakers' statements, not private information or harmful messages. Instead, we expect that people will understand each other if they recognize the possibility of different interpretations of their language.

## Acknowledgements

We would like to thank Mr. Aoto Tsuchiya at Keio University for the fundamental analysis of the architecture.

This work was supported by JST CREST Grant number JPMJCR19A1, Japan.

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

## A  Other prompt templates

We tested the following prompt templates to extract important statements:

- `Select the two most important statements from the dialogue and write their numbers`: this prompt template was used for the uniqueness evaluation.

- `Write down the two most important statements from the dialogue`: when we tested this prompt template, the LLM did not always write down the statements accurately, which made the analysis difficult.

- `Which statements are important?`: when we tested this prompt template, the format of the output was unstable, which made the analysis difficult.

Note that most statements in JPersonaChat contain some kind of information about the speaker's personal attributes. Therefore, it is difficult to subjectively judge whether important statements are accurately extracted.

## B  Qualitative results

Tables 9, 10, 11, 12, and 13 show five example dialogues with the lowest similarity scores. Although there may be more than one pair of SCAINs in a dialogue, we only show the SCAINs with the lowest similarity score in the dialogue. We also

show the rephrased statements from the complete and omitted dialogues, as well as the similarity scores between the rephrased statements. The dialogues and rephrased statements were originally in Japanese and were translated into English for inclusion in this paper.

## C Codes

The codes used to extract the SCAINs and analyze the results are available at https://github.com/imai-laboratory/summary_scain.

Table 9: Example dialogue 1. *Italic text* represents the SCAINs. Underlined text represents the core statement following the SCAINs. **Bold text** represents the important statements.

| | |
|---|---|
| A: | Hey, can I talk to you for a second? |
| B: | What's up? |
| A: | **My boyfriend hates my Kansai dialect. He says I sound like an old lady.** |
| B: | I live in the Kinki region, so I don't agree at all, but that's a bit harsh! |
| A: | Definitely! But I'm a high school graduate, and there aren't many men who will go out with me because I'm moody. |
| B: | **I don't think that matters. I'm a college graduate, but one of my classmates married a high school graduate. Anyway, cheer up!** |
| A: | Thanks! Oh, by the way, I just remembered something even worse that was said to me! |
| B: | Even worse! |
| A: | When I wore a leopard print tank top to match my brown hair on a date, he asked me if I was a leopard. |
| B: | *Well, I don't know if I can agree with that. But he's not so bad for a guy who can make a joke, is he?* |
| A: | *Really? Is he? Do you think we'll continue to get along as a couple?* |
| B: | Maybe you're a surprisingly good match! |
| A: | I see! That's a relief. I'm glad I talked to you about it! |

| | |
|---|---|
| Rephrased statement from the complete dialogue | You guys understand each other well and respect each other so I am sure you will get along together. |
| Rephrased statement from the omitted dialogue | Maybe the leopard print tank top to match your brown hair would look good! |
| Similarity score | 0.775 |

Table 10: Example dialogue 2. *Italic text* represents the SCAINs. Underlined text represents the core statement following the SCAINs. **Bold text** represents the important statements.

| | |
|---|---|
| A: | Hello. What are you doing these days? |
| B: | Lately, I've been working a lot at my current company, which I joined as a graduate hire. Because of the work, I'm always short of sleep. How about you? |
| A: | **It's tough being an employee. After changing jobs many times, I've found that being a freelancer suits me better.** |
| B: | Freelance. That's cool. I'm only good at being patient, so I'm going to stay with my current company for a while. |
| A: | You are respectable. I'm the kind of person who makes enemies. I'm sure that's my problem. |
| B: | Do you have that harsh a personality? |
| A: | Well, I think I'm good at taking the lead, but many people seem to think I'm too forward for a college dropout. |
| B: | ***That's a bit of prejudice. But there are people like that everywhere. I'm glad you found something that fits your personality!*** |
| A: | *Thank you! That makes me feel better. By the way, have you been traveling lately?* |
| B: | Not at all. Have you? |
| A: | Every day is like a trip for me. I travel all over Japan without settling down anywhere. |
| B: | I'd like to live like that just once! |

| | |
|---|---|
| Rephrased statement from the complete dialogue | I've been so busy with work lately that I haven't had time to travel. Have you traveled lately? |
| Rephrased statement from the omitted dialogue | I think I am good at taking the lead, but actually, I am still a work in progress and have many areas for improvement. |
| Similarity score | 0.778 |

Table 11: Example dialogue 3. *Italic text* represents the SCAINs. Underlined text represents the core statement following the SCAINs. **Bold text** represents the important statements.

| | |
|---|---|
| A: | Hello. How are you? |
| B: | I've been well. How about you? |
| A: | I have been well too. On my days off, I enjoy my hobby, photography. |
| B: | **Photography is nice. I live near the sea, so I often go for a walk.** |
| A: | So you live near the sea. It must be beautiful. I live near the mountains. |
| B: | I see. I like mountains, but I used to live in Okinawa and I really wanted to live near the sea. |
| A: | You feel at home in a place similar to where you were born and raised. I am from Aomori, so I understand how you feel. |
| B: | **That's right. I also tend to get angry, so seeing the ocean is good for calming me down.** |
| A: | Nature is very calming, isn't it? I want to go take pictures of the seaside next time. |
| B: | Yes, I think the sea is also very nice because it has many different faces. |
| A: | Please let me know if you have any other recommendations for places to go. |
| B: | Sure. By the way, I am good at dancing, how about you? |
| A: | *I have hardly ever danced. I like to watch dancing, though.* |
| B: | *I see. But if you like to watch dancing, I will show you next time.* |
| A: | I would love to. |

| | |
|---|---|
| Rephrased statement from the complete dialogue | Could you please let me know if there are any other recommended seaside photo spots? Thank you very much in advance. |
| Rephrased statement from the omitted dialogue | Let's dance together sometime. I like dancing too, so I am looking forward to it. |
| Similarity score | 0.783 |

Table 12: Example dialogue 4. *Italic text* represents the SCAINs. Underlined text represents the core statement following the SCAINs. **Bold text** represents the important statements.

| | |
|---|---|
| A: | Nice to meet you. Hello. |
| B: | Hello. I'm a member of the track team at my high school. |
| A: | So you are a high school student! I am an esthetician. |
| B: | **That's cool. Actually, I'm not good at getting up early, so morning practice is hard for me.** |
| A: | I feel for you. When I was a club member, it was hard for me to get up early, too. |
| B: | I see. How did you manage to get up in the morning? |
| A: | I set many alarms. My mother used to wake me up too. |
| B: | I live with my grandmother, so she wakes me up, but she wakes me up too softly. |
| A: | *I can imagine. By the way, my favorite author is Kenji Miyazawa. What books do you like to read?* |
| B: | *I don't read many books, but I know one of his books. It's a fantasy, isn't it?* |
| A: | You're right. Please try reading it. |
| B: | Okay. I'll look for it. |
| A: | **By all means! I'm also working hard to become a manicurist, so please do your best in your club activities.** |

| | |
|---|---|
| Rephrased statement from the complete dialogue | Kenji Miyazawa's "Night on the Galactic Railroad" is a very moving fantasy novel. Please read it. |
| Rephrased statement from the omitted dialogue | I used to have trouble getting up in the morning too, but I had to set several alarms and even had my mother wake me up. It's sweet of your grandmother to wake you up, but often she wakes you up too softly. If that's the case, please try the alarm app I use. It may be an effective way to wake up. |
| Similarity score | 0.784 |

Table 13: Example dialogue 5. *Italic text* represents the SCAINs. Underlined text represents the core statement following the SCAINs. **Bold text** represents the important statements.

| | |
|---|---|
| A: | Long time no see. How is your work going? |
| B: | **Well, I'm on a leave of absence. I have the problem that I don't have enough money!** |
| A: | You don't have much money saved? I quit my job and now I don't have a job, but I did save some money, so I guess I'll be okay for now. |
| B: | *I envy you! I don't have any savings at all, and since I took a leave of absence, I can't even eat my favorite sushi, I feel like crying.* |
| A: | *That's too bad. Next time you pick out an outfit that looks good on me with short hair, I'll treat you!* |
| B: | Really? I'll pick one! I have dark hair, so I'm often told I look good in Gothic Lolita, but what kind of clothes do you like? |
| A: | I don't think I'm Gothic Lolita at least. But I do like skirts, so I'll go with a skirt or dress that goes with short hair! |
| B: | Wow, one-piece dresses are cute! Do you think short hair looks good on me, even though I have a lot of hair? |
| A: | I'm a sociable person, but I'm not a flatterer and I don't lie! |
| B: | Then I believe you! When shall we go? |
| A: | Anytime! I don't have a job. |
| B: | Yeah, yeah, we're both unemployed! |

| | |
|---|---|
| Rephrased statement from the complete dialogue | What kind of clothes do you like to wear? People often say I look good in Gothic Lolita because I have dark hair, but what styles do you think suit you? |
| Rephrased statement from the omitted dialogue | "Yes, but it's tough worrying about money. I'm unemployed right now, but I did save some money, so I'm relieved for now." |
| Similarity score | 0.785 |