# OpenReview forum: "Identifying Statements Crucial for Awareness of Interpretive Nonsense to Prevent Communication Breakdowns"
_EMNLP/2023/Conference — EMNLP 2023 Main_

### Official Review · Reviewer_e24W · 2023-07-21

**Soundness:** 4

**Excitement:**

4: Strong: This paper deepens the understanding of some phenomenon or lowers the barriers to an existing research direction.

**Paper Topic And Main Contributions:**

The authors introduce the concept of Statements Crucial for Awareness of Interpretive Nonsense (SCAINs) which are missed statements that lead to misunderstandings later in a dialogue. To determine which are these crucial statements they simulate missing information and generate texts with/without the missing information. SCAINs are not only important statements, but statements that may impact the interpretation.

**Questions For The Authors:**

- page 3 "we compute the similarity score between 2 vectors " - which vectors do you mean? rephrased and original? candidate vs core?
- The prompt for 'important statements' seems weak/unclear (what is an important statement) - were multiple prompt templates tested?
-  are there limitations in the quality of gpt 3.5-turbo for japanese?
- Would it be possible for the LLM to guess the correct interpretation in the omitted scenario even it was actually a SCAIN that was missing? For example because the correct interpretation is the 'regular' interpretation and the LLM did not pick up on the alternative interpretation from the dialogue?

**Reasons To Accept:**

- very interesting and unique topic that can be very relevant also for quality of summarization tasks
- Clear explanation of the approach with mostly sufficient details for reproduction
- Good reflection on limitations.

**Reasons To Reject:**

- now only short distance SCAINS are considered (pair of consecutive statements), longer term SCAINs may be interesting as well, which the authors mention as a limitation as well
- how was the threshold determined - only "on the basis of the observation of the rephrased statements" but this is a bit vague - which is mentioned in the limitations as well
- As I understand it, the LLMs determine which are SCAINs (below 0.85) and these vary depending on the prompt (validity test: 65 SCAINs and uniqueness 125 SCAINs) for the same set. So what's the 'ground truth' then? Or is this difference because of the pairs (but why is this different in the uniqueness condition then)


**Reproducibility:**

3: Could reproduce the results with some difficulty. The settings of parameters are underspecified or subjectively determined; the training/evaluation data are not widely available.

**Reviewer Confidence:**

3: Pretty sure, but there's a chance I missed something. Although I have a good feel for this area in general, I did not carefully check the paper's details, e.g., the math, experimental design, or novelty.

---

> ### Author Rebuttal · Authors · 2023-08-28
>
> Thank you for insightful comments and questions. We will address them comprehensively.
>
> > now only short distance SCAINS are considered (pair of consecutive statements), longer term SCAINs may be interesting as well, which the authors mention as a limitation as well
>
> As you pointed out, the current method identifies SCAINs from the statements immediately preceding the core statement. We recognize that there is a need to identify more distant SCAINs. The current study aimed to establish the concept of SCAIN, so we defined the candidate statements as the immediately preceding statements for the first step. We will extend the method to identify more distant SCAINs as an important future work. In addition, there will be an issue of computational cost as we increase the number of patterns of candidate statements. We will be able to predict the change in interpretation of the core statement caused by the omission of more distant statements if we provide sufficient computational resources. This extension will be done when the inference process of the LLM becomes more efficient thanks to research progress.
>
> We will add this explanation of our perspective to the limitation section.
>
> > how was the threshold determined - only "on the basis of the observation of the rephrased statements" but this is a bit vague - which is mentioned in the limitations as well
>
> As you pointed out, we predetermined the threshold based on the observation of the rephrased statements in the current method. However, we will be able to determine a strict threshold by analyzing the relationship between the similarity score and the interpretation of the core statement in more detail.
>
> > As I understand it, the LLMs determine which are SCAINs (below 0.85) and these vary depending on the prompt (validity test: 65 SCAINs and uniqueness 125 SCAINs) for the same set. So what's the 'ground truth' then? Or is this difference because of the pairs (but why is this different in the uniqueness condition then)
>
> We apologize for the lack of explanation. The SCAINs were identified using the same procedure for both validity and uniqueness evaluation. However, the ways of counting the SCAINs are different between the validity and the uniqueness evaluation. During the validity evaluation, we counted the number of pairs of SCAINs and identified 65 pairs. We considered the SCAINs as pairs because, for the validity evaluation, we wanted to analyze the tendency of the dialogues containing the SCAINs. On the other hand, during the uniqueness evaluation, we counted the number of SCAINs and identified 125 SCAINs. We considered the SCAINs as individual statements because, for the uniqueness evaluation, we wanted to analyze the relationship between the classifications of the statements in terms of being SCAINs and important statements. The number of SCAINs is less than twice the number of pairs because there are five overlapping pairs. We will add a more detailed explanation.
>
> > page 3 "we compute the similarity score between 2 vectors " - which vectors do you mean? rephrased and original? candidate vs core?
>
> We apologize for the lack of explanation. We compared how similar the embedding vector of the rephrased statement is when using the complete dialogue prompt versus when using the omitted dialogue prompt. We will add a more detailed explanation.
>
> > The prompt for 'important statements' seems weak/unclear (what is an important statement) - were multiple prompt templates tested?
>
> We tested the following prompt templates to extract important statements:
>
> 1. Select the two most important statements from the dialogue and write their numbers (used for uniqueness evaluation).
>
> 2. Write down the two most important statements from the dialogue (the LLM did not always write down the statements accurately, which made the analysis difficult).
>
> 3. Which statements are important? (the format of the output was unstable, which made the analysis difficult).
>
> Note that most statements in JPersonaChat contain some kind of information about the speaker's personal attributes. Therefore, it is difficult to subjectively judge whether important statements are accurately extracted. We will add this explanation of prompt templates as an appendix.
>
> > are there limitations in the quality of gpt 3.5-turbo for japanese?
>
> As shown in the following report, the performance of GPT-4 in Japanese language processing is inferior to that of English. We speculate that GPT-3.5-turbo has a similar tendency. However, we have confirmed that GPT-3.5-turbo shows sufficient accuracy for the task of rephrasing a sentence into a more specific one. We will add this explanation to the Limitations section.
>
> * OpenAI. 2023. GPT-4 technical report.
>
> > Would it be possible for the LLM to guess the correct interpretation in the omitted scenario even it was actually a SCAIN that was missing? For example because the correct interpretation is the 'regular' interpretation and the LLM did not pick up on the alternative interpretation from the dialogue?
>
> When SCAINs are omitted from the dialogue, the interpretation of the core statement changes significantly, making it difficult to predict the primary intent of the core statement. If the LLM is informed that some statements have been omitted, it can generate a possible interpretation of the core statement by predicting the content of the omitted statements. However, the generated interpretation is likely to be different from the primary intent because crucial information for interpreting the core statement has been lost. We will add this concern to the discussion section.

---

### Official Review · Reviewer_qqvW · 2023-08-04

**Soundness:** 2

**Excitement:**

2: Mediocre: This paper makes marginal contributions (vs non-contemporaneous work), so I would rather not see it in the conference.

**Paper Topic And Main Contributions:**


The aim of this paper is to identify statements in a dialogue such that if a listener missed them, the interpretation of
some subsequent utterances might change. They call these statements "SCAINs", explained in the title. A simple example
of such a thing would be a sentence introducing a referent for a pronoun which was missed out, forcing the pronoun to
refer to some eligible entity earlier in the conversation.  The title of the paper mentions "interpretive nonsense" but
all the examples are more to do with contextual effects on locally ambiguous utterances rather than sense/nonsense.

The paper proposes a method of locating such statements with the eventual aim of incorporating a mechanism into a
conversational support system of some kind.

To do this they take a dialogue from an existing dataset and omit a statement at position j. This gives two versions of
the dialogue. Then an LLM is prompted to paraphrase the statement at the position j+1 in both versions, one with the
original j position statement present and one without. Next the embeddings of the two paraphrased statements are
compared and if they are sufficiently far apart a SCAIN is hypothesised.

Finally, human raters are used to validate the decision. Raters are also asked to judge the importance of the various
statements, and the results show that importance is a distinct dimension to the property of being a SCAIN.


**Reasons To Accept:**

The paper is clearly written and the method for locating SCAINs might have applications elsewhere, such as in assembling a corpus of contextually-dependent ambiguities.

**Reasons To Reject:**

The paper lacks any convincing motivation for the need for such a system. It sounds like a rather artificial problem, a perception enhanced by the weirdness of the dialogues used for illustration. Even allowing for translation effects, they sound like two bots talking, and on following up the references I found that they are indeed completely artificial.

**Reproducibility:**

2: Would be hard pressed to reproduce the results. The contribution depends on data that are simply not available outside the author's institution or consortium; not enough details are provided.

**Reviewer Confidence:**

3: Pretty sure, but there's a chance I missed something. Although I have a good feel for this area in general, I did not carefully check the paper's details, e.g., the math, experimental design, or novelty.

---

> ### Author Rebuttal · Authors · 2023-08-28
>
> Thank you for reviewing our paper and giving insightful comments. We will explain our motivation in detail.
>
> > They call these statements "SCAINs", explained in the title. A simple example of such a thing would be a sentence introducing a referent for a pronoun which was missed out, forcing the pronoun to refer to some eligible entity earlier in the conversation. The title of the paper mentions "interpretive nonsense" but all the examples are more to do with contextual effects on locally ambiguous utterances rather than sense/nonsense.
>
> The "nonsense" abbreviated in SCAIN has a special meaning. A listener who has not heard SCAINs may interpret a subsequent statement differently from the speaker's primary intent. However, the listener is likely to believe that his/her interpretation is correct. The SCAIN extractor computes the change in interpretation by a listener who missed hearing the candidate statements. In this sense, SCAINs are different from locally ambiguous utterances. Even if a statement is locally ambiguous, the statement is not a SCAIN unless the interpretation changes due to mishearing. The SCAIN extractor picks up the statements when mishearing may change the interpretation of the subsequent statement, but the listener is not aware of the change, so the conversation support system will alert the speakers to pay attention to the different interpretations. We will add this explanation of our motivation to the Introduction section.
>
> > The paper lacks any convincing motivation for the need for such a system.
>
> We assume that the conversation support system will present SCAINs to the speakers in a conversation, not to the listener who has misinterpreted the subsequent statement. The system is designed to prevent the speaker from continuing a one-way conversation when the speaker's utterance has been misinterpreted. When presented with the SCAINs, the speaker will pay close attention to the verbal and nonverbal responses of the listener. The speaker will be able to identify the communication breakdowns earlier and recover from the breakdowns by supplementing the SCAINs with more information. The system will prevent such situations and let the speakers be aware of communication breakdowns.
>
> Note that the SCAINs do not indicate that there are communication breakdowns. Instead, if a listener did not hear the SCAINs, there would be communication breakdowns. Therefore, the system is designed to alert speakers to the possibility of communication breakdowns, not to detect actual communication breakdowns.
>
> We will add these details of the application system to the discussion section.
>
> > It sounds like a rather artificial problem, a perception enhanced by the weirdness of the dialogues used for illustration. Even allowing for translation effects, they sound like two bots talking, and on following up the references I found that they are indeed completely artificial.
>
> As you pointed out, the dialogue for illustration was created artificially. However, communication breakdowns due to mishearing can occur in a real environment. There is no existing study that represents the change in interpretation caused by missing statements in the format of natural language. We believe that the significance of our paper lies in proposing the concept of SCAIN as an idea that can be useful for communication support.

---

### Official Review · Reviewer_g4QE · 2023-08-12

**Typos Grammar Style And Presentation Improvements:** Line 295
**Soundness:** 3

**Excitement:**

3: Ambivalent: It has merits (e.g., it reports state-of-the-art results, the idea is nice), but there are key weaknesses (e.g., it describes incremental work), and it can significantly benefit from another round of revision. However, I won't object to accepting it if my co-reviewers champion it.

**Missing References:**

Huiyuan Xie, Zhenghao Liu, Chenyan Xiong, Zhiyuan Liu, and Ann Copestake. 2021. TIAGE: A Benchmark for Topic-Shift Aware Dialog Modeling. In Findings of the Association for Computational Linguistics: EMNLP 2021, pages 1684–1690, Punta Cana, Dominican Republic. Association for Computational Linguistics.

**Paper Topic And Main Contributions:**

This paper studies the communication breakdowns problem which often occurs when a listener misses certain statements during online conversation interactions. Authors define the statements crucial for interpreting other statements as SCAINs (Statements Crucial for Awareness of Interpretive Nonsense). The contributions of it are two-fold: 1) proposes the method of extracting SCAINs. 2) validates that the SCAINs are crucial to the interpretation of dialogues and shows the relations and difference between SCAINs and important statements.

**Questions For The Authors:**

1) Please illustrate details of applying the extractor in online conversation tools. When communication breakdowns occur, does the extractor have access to the complete dialogues and who will the extractor notify? senders and listeners? It would be much better if a detailed example could be included.

2) I am curious about the difference between SCAINs and utterances eliciting new topics (where a topic shift occurs).

**Reasons To Accept:**

1) This work defines a new kind of utterance related to the interpretation of other statements, which can indicate whether a communication breakdown occurs within the current dialogue.

2) The experimental results and statistical analysis are comprehensive and clear.

**Reasons To Reject:**

1) The work relies on LLM too much and the computational efficiency seems not high. Firstly, for each turn of dialogues, the LLM is used to paraphrase two core statements with two forward passes. Then the embedding model is employed to obtain the embeddings of the two core statements, two forward passes too. Four forward passes of LLM are required for each turn of dialogues.

2) Some experimental results seem not much significant and convincing. E.g., In line 264, the SCAINs condition is defined as the similarity being lower than 0.85 but the Non-SCAINs condition is defined as 0.9897. While the inconsistent of condition threshold is explained by authors: obtaining the same fraction of the statements as SCAINs, it can be replaced by randomly sampled from the statements with the similarity being higher than 0.85, at the meanwhile, keeping the fraction of statements the same.

**Reproducibility:**

4: Could mostly reproduce the results, but there may be some variation because of sample variance or minor variations in their interpretation of the protocol or method.

**Reviewer Confidence:**

3: Pretty sure, but there's a chance I missed something. Although I have a good feel for this area in general, I did not carefully check the paper's details, e.g., the math, experimental design, or novelty.

---

> ### Author Rebuttal · Authors · 2023-08-28
>
> Thank you for the insightful comments and questions. We will address them comprehensively.
>
> > The work relies on LLM too much and the computational efficiency seems not high. Firstly, for each turn of dialogues, the LLM is used to paraphrase two core statements with two forward passes. Then the embedding model is employed to obtain the embeddings of the two core statements, two forward passes too. Four forward passes of LLM are required for each turn of dialogues.
>
> As you pointed out, the SCAIN extractor requires four forward passes of the LLM for each utterance. Currently, the computation time of the LLM cannot be ignored. However, the inference process of the LLM is becoming more efficient thanks to the progress in computer hardware and LLM research. Therefore, we expect that the SCAINs will be extracted in a much shorter time for real-time applications. We will add this limitation of the current method to the limitation section.
>
> > Some experimental results seem not much significant and convincing. E.g., In line 264, the SCAINs condition is defined as the similarity being lower than 0.85 but the Non-SCAINs condition is defined as 0.9897. While the inconsistent of condition threshold is explained by authors: obtaining the same fraction of the statements as SCAINs, it can be replaced by randomly sampled from the statements with the similarity being higher than 0.85, at the meanwhile, keeping the fraction of statements the same.
>
> As you pointed out, the results of the validity experiment do not guarantee that the predetermined value of the threshold is valid. We expect dialogues with intermediate similarities between the SCAINs and Non-SCAINs conditions will fall into intermediate results. However, it is important to set the extreme conditions for the basic analysis. The evaluation results show that the proposed method can extract dialogues with significantly different tendencies, at least for extreme conditions. We will add explanations to clarify that the paper describes the basic analysis to find out the tendencies with the extreme conditions.
>
> > Please illustrate details of applying the extractor in online conversation tools. When communication breakdowns occur, does the extractor have access to the complete dialogues and who will the extractor notify? senders and listeners? It would be much better if a detailed example could be included.
>
> We apologize for the lack of explanation. The SCAINs do not indicate that there are communication breakdowns. Instead, if a listener missed hearing the SCAINs, there would be communication breakdowns. Therefore, the tool is designed to alert speakers to the possibility of communication breakdowns, not to detect actual communication breakdowns.
>
> The tool will be useful for improving interaction between speakers. We assume that the tool will inform the sender of SCAINs which statements are SCAINs during conversations. Note that the SCAINs only indicate the possibility of communication breakdowns, so it is up to the sender how to use this information. Because the SCAINs are presented to the sender, the sender will pay close attention to verbal and nonverbal responses from the listener. The sender will be able to identify the communication breakdowns earlier and recover from the breakdowns by supplementing the SCAINs with more information. If the speakers do not use the tool, the sender will not care if his/her intent is conveyed to the listener and will continue with a one-way conversation. The tool will prevent such situations and let the speakers be aware of communication breakdowns.
>
> We will add these details of the application tool to the discussion section.
>
> > I am curious about the difference between SCAINs and utterances eliciting new topics (where a topic shift occurs).
>
> Thank you for the suggestion. We tested whether the SCAINs predict topic shifts using the TIAGE dataset. As a result, there is no significant relationship between the SCAINs and topic shifts. We show the cross-table and the results of a Chi-square test.
>
> |    |Topic shift|Non-topic shift|Total|
> |----|----|----|----|
> |SCAIN|18|66|84|
> |Non-SCAIN|297|1083|1380|
> |Total|315|1149|1464|
>
> $\chi ^2 (1) = 0.0$, $p=1.0$.
>
> We speculate that there is no significant relationship between the SCAINs and topic shifts because the topic shifts do not necessarily change the interpretation of the subsequent statement. Even if there is a topic shift, the listener may correctly interpret a statement that contains sufficient information about the new topic. We will add this evaluation and discussion to the appropriate sections.
>
> > Missing References:
>
> > Huiyuan Xie, Zhenghao Liu, Chenyan Xiong, Zhiyuan Liu, and Ann Copestake. 2021. TIAGE: A Benchmark for Topic-Shift Aware Dialog Modeling. In Findings of the Association for Computational Linguistics: EMNLP 2021, pages 1684–1690, Punta Cana, Dominican Republic. Association for Computational Linguistics.
>
> Based on your suggestion, we will add a reference to the paper on topic-shift aware dialog modeling and expand the Related Work section to present research in this area.
>
> > Line 295: The second “omitted statements” should be “complete statements”.
>
> We apologize for the confusion. We will revise the sentence to make it clearer.
>
> Before revision:
>
> At this point, we asked the raters not to change the answers for the omitted dialogue to ensure that they were rating the omitted dialogue without reading the **omitted statements**.
>
> After revision:
>
> At this point, we asked the raters not to change the answers for the omitted dialogue to ensure that they were rating the omitted dialogue without reading the **complete dialogue**.

---

### Meta-Review · Area_Chair_qXxE · 2023-09-18

**Recommendation:** 3

**Metareview:**

This paper introduces the task of identifying missed statements that lead to misunderstandings later in a dialogue. This is an interesting task; the proposed solution is novel and might have applications in other tasks. The paper is well-written, and the experiments are comprehensive. The problem statement needs further elaboration to provide convincing motivation for the need for such a system. The practical applications must be clarified.

The paper will benefit from all the modifications mentioned in the authors' rebuttals. Specifically, the authors need to add:

- Better motivation and problem statement, along with examples
- Clarifications on the threshold setting
- Discussions on the computational cost
- Explaining the differences between validity and uniqueness evaluations

---

### Decision · Program_Chairs · 2023-10-07

**Decision:**

Accept-Main

**Comment:**

This paper introduces the task of identifying missed statements that lead to misunderstandings later in a dialogue. This is an interesting task; the proposed solution is novel and might have applications in other tasks. The paper is well-written, and the experiments are comprehensive. The problem statement needs further elaboration to provide convincing motivation for the need for such a system. The practical applications must be clarified.

The paper will benefit from all the modifications mentioned in the authors' rebuttals. Specifically, the authors need to add:

- Better motivation and problem statement, along with examples
- Clarifications on the threshold setting
- Discussions on the computational cost
- Explaining the differences between validity and uniqueness evaluations